

# Non-host class II ribonucleotide reductase in Thermus viruses: sequence adaptation and host interaction

Christoph Loderer[1], Karin Holmfeldt[2] and Daniel Lundin[2,3]

[1] Institute for Microbiology, Technische Universität Dresden, Dresden, Saxony, Germany
[2] Centre for Ecology and Evolution in Microbial model Systems—EEMiS, Linnaeus University, Kalmar, Sweden
[3] Department of Biochemistry and Biophysics, Stockholm University, Stockholm, Sweden

## ABSTRACT

Ribonucleotide reductases (RNR) are essential enzymes for all known life forms. Their current taxonomic distribution suggests extensive horizontal gene transfer e.g., by processes involving viruses. To improve our understanding of the underlying processes, we characterized a monomeric class II RNR (NrdJm) enzyme from a Thermus virus, a subclass not present in any sequenced *Thermus* spp. genome. Phylogenetic analysis revealed a distant origin of the *nrdJm* gene with the most closely related sequences found in mesophiles or moderate thermophiles from the Firmicutes phylum. GC-content, codon usage and the ratio of coding to non-coding substitutions (dN/dS) suggest extensive adaptation of the gene in the virus in terms of nucleotide composition and amino acid sequence. The NrdJm enzyme is a monomeric $B_{12}$-dependent RNR with nucleoside triphosphate specificity. It exhibits a temperature optimum at 60–70 °C, which is in the range of the growth optimum of *Thermus* spp. Experiments in combination with the *Thermus thermophilus* thioredoxin system show that the enzyme is able to retrieve electrons from the host NADPH pool via host thioredoxin and thioredoxin reductases. This is different from other characterized viral RNRs such as T4 phage RNR, where a viral thioredoxin is present. We hence show that the monomeric class II RNR, present in Thermus viruses, was likely transferred from an organism phylogenetically distant from the one they were isolated from, and adapted to the new host in genetic signature and amino acids sequence.

Corresponding author
Christoph Loderer,
christoph.loderer@tu-dresden.de

# INTRODUCTION

Ribonucleotide reductases (RNRs) catalyze the reduction of ribonucleotides to the corresponding deoxyribonucleotides. Since this reaction is the only known biological *de novo* synthesis pathway for deoxyribonucleotides, it is essential to all known forms of life (*Hofer et al., 2012*; *Lundin et al., 2015*; *Nordlund & Reichard, 2006*). All known RNRs share a common reaction mechanism based on a catalytic thiyl radical and a common 10 stranded α/β fold. There are three different classes of RNRs that differ in features like radical generation mechanism, oligomeric state and terminal reductant (*Hofer et al., 2012*;

*Lundin et al., 2015*; *Nordlund & Reichard, 2006*; *Torrents, 2014*). Class I requires oxygen and is made up of a heterodimer consisting of an NrdA (catalytic subunit) homodimer and an NrdB (radical generating subunit) homodimer. Class II is a single component NrdJ protein that is vitamin $B_{12}$-dependent. Most NrdJs are homodimeric, but the NrdJm subclass is monomeric. Class III is oxygen sensitive, both by the fact that oxygen cleaves the catalytic NrdD component in activated state and by the fact that the radical SAM family activase, NrdG, is oxygen sensitive (*Lundin et al., 2015*; *Nordlund & Reichard, 2006*). The organismal distribution of the different classes of RNRs does not follow the phylogeny of organisms but rather their biochemical requirements. Phylogenetic analysis suggests that multiple events of horizontal gene transfer (HGT), at phylogenetic distances varying from close relatives to inter domain transfers, are required to explain the distribution of RNRs (*Lundin et al., 2010*).

HGT can be mediated by viruses via transduction (*Soucy, Huang & Gogarten, 2015*). Indications of transduction-mediated interdomain HGT have been described for RNR in which a trio of sequences most closely related, phylogenetically, were found in a halophilic bacterium (*Salinibacter ruber*), a halophilic archaeon (*Natronomonas pharaonis*) and a virus isolated from a saltern brine (Halophage AAJ-2005) (*Lundin et al., 2010*). However, to cover large phylogenetic distances, such as between phyla or domains, the process likely requires considerable evolution of host specificity in the virus, a process that is poorly known and mostly studied at relatively short phylogenetic distances (*Koskella & Meaden, 2013*). HGT by transduction is a stepwise process in which the transferred genes need to survive sufficiently long in the virus genome to come in contact with the receiving cellular organism during infection. In cases where genes transfer between distantly related cellular organisms, genes must remain in the virus during evolution of host specificity, suggesting that selection for a function provided by the transferred genes is likely to play a role. During their tenure in the viral genome, the transferred genes may also see selection to acquire a suitable GC content, codon usage and potential other genetic signatures. Ultimately, the transferred gene needs to either provide a selective advantage to its new host or by selfish processes ensure selection (*Soucy, Huang & Gogarten, 2015*). RNR is well suited to study these processes by being an essential enzyme—as it is required for DNA replication and repair—but is not an enzyme that is physically associated with a large multicomponent complex. Moreover, RNRs can be found in many double stranded DNA (dsDNA) viruses (*Dwivedi et al., 2013*; *Sakowski et al., 2014*) that can potentially act as vectors for transduction.

For a transferred RNR to survive the stepwise process from the moment of transfer to potential selective advantage, the foremost functional requirement is the presence of a compatible reducing system in the receiving organism or in the infected host in the case of viruses. Three different systems to reduce the enzyme after catalysis have been found. In characterized class I and II, the reductant is thioredoxins or glutaredoxins (*Booker & Stubbe, 1993*; *Holmgren & Sengupta, 2010*), whereas in class III thioredoxin has been identified as the reductant in one enzyme (*Wei et al., 2015*) and formate in another (*Mulliez et al., 1995*). In the case of the *Escherichia* virus T4 (T4) NrdAB, the phage delivers a separate T4-thioredoxin that interacts with the native *E. coli* thioredoxin reductase (*Berglund &*

*Holmgren, 1975*; *Tseng et al., 1990*). Other viruses, such as human *Herpes simplex* virus (HSV), do not encode viral thioredoxins and thus rely on their host's reduction system (*Fawl & Roizman, 1994*).

Adaptation of GC content and codon usage is an important aspect of genetic integration and genome evolution. This is true for RNRs not only in that the gene's coding sequence needs to adapt to overall GC content, as for all transferred genes, but also because RNR regulates the levels of the four dNTPs in cells (*Hofer et al., 2012*; *Nordlund & Reichard, 2006*). This is performed by allosteric substrate specificity regulation under the control of concentrations of individual dNTPs. In short, binding of a nucleotide –ATP, dATP, dCTP, dTTP and dGTP –to a specificity site, affects affinity for different nucleotides in the substrate binding pocket via the so called ''loop 2'' (*Larsson et al., 2004*). Imbalances in the dNTP pools are mutagenic (*Hofer et al., 2012*; *Mathews, 2014*) and can hence potentially lead to the demise of a genetic lineage in extreme cases.

Most viral RNRs are similar to one found in their host organisms. For instance, T4-NrdA amino acid sequence is 54% identical to *Escherichia coli* NrdA. Thus, many viruses may have retrieved their RNRs from their corresponding host genomes. But there are exceptions. Screening the database RNRdb (http://rnrdb.pfitmap.org) for RNRs in viruses with no corresponding gene in the presumed host, we found three Thermus viruses (TV) with *nrdJm* genes, encoding monomeric class II RNRs, not present in their hosts. By analyzing the phylogenetic history and sequence signatures of these genes, we traced back the evolutionary origin and investigated the process of genetic adaptation. In a biochemical characterization we investigated the functionality of the enzyme with respect to native conditions as well as its interaction with the host thioredoxin system.

## MATERIALS & METHODS

### Bioinformatics

All NrdJm amino acid sequences from NCBI's RefSeq database were downloaded from RNRdb (http://rnrdb.pfitmap.org) and clustered with USEARCH (*Edgar, 2010*) at 90% sequence identity to reduce redundancy. Non full-length sequences and sequences of dubious quality were manually removed, before aligning all 364 sequences with ProbCons (*Do et al., 2005*). Trustworthy alignment positions were selected with the BMGE algorithm (*Criscuolo & Gribaldo, 2010*) using the BLOSUM30 substitution matrix, ending up with 363 well-aligned positions forming 350 distinct alignment patterns. RAxML version 8.2.4 (*Stamatakis, 2014*) was used to estimate a phylogeny, using the PROTGAMMAAUTO model, rapid bootstrapping with the autoMRE bootstopping followed by a full maximum likelihood tree search. The phylogeny is available at Figshare (https://doi.org/10.17045/sthlmuni.7117430.v1).

To search for sequences from metagenomes, we downloaded all TARA Ocean ORFs (https://doi.org/10.6084/m9.figshare.4902917, (*Delmont et al., 2018*)), all ORFs from the Human Microbiome Project (2017-01-09; (*Human Microbiome Project C, 2012a*; *Human Microbiome Project C, 2012b*)) the majority of archaeal and bacterial metagenome assembled genomes (MAGs) and single amplified genomes (SAGs) from IMG/MER
(4910 MAGs, 2230 SAGs) plus 53 aquatic and soil metagenomes, in particular those with project names containing "virus", "phage", "therm" or "hot" (SI) (*Markowitz et al., 2008*). Together, we downloaded a total of 250,881,638 ORFs. We used hmm profiles designed for each clan in the phylogeny to search the sequences. We found 181 sequences with a best match to the profile designed from the TV clan. These were aligned to the original alignment using Clustal Omega in profile mode (*Sievers et al., 2011*) and phylogenetically placed in the full phylogeny with RAxML (*Stamatakis, 2014*) (https://doi.org/10.17045/sthlmuni.7642343.v1).

A likelihood ratio test of significant overrepresentation of non-synonymous to silent (dN/dS) was estimated with codonml from PAML (*Stamatakis, 2014*; *Yang, 1997*; *Yang, 2007*) by running the program with a fixed and free dN/dS respectively, and the branch leading to the two Thermus virus sequences (marked with "#1" in Fig. 1B) designated as the "foreground" branch in the free dN/dS run. This analysis was performed on the subtree in Fig. 1B, using the same alignment as for the full protein sequence tree, reverse translated into the correct nucleotide sequence for each protein sequence. A $\chi^2$ *p*-value of the likelihood ratio test was calculated with one degree of freedom. We could not find correct gene sequences for seven taxa (RefSeq accession numbers: WP_102410887, WP_033167051, WP_054955013, WP_065068364, WP_088370373, WP_087372021, WP_093315575), so they were left out of the analysis (https://doi.org/10.17045/sthlmuni.7642463.v1).

The similarity of codon usage as euclidean distance between the codon frequencies was calculated as described elsewhere (*Popa, Landan & Dagan, 2017*).

The model of the TV P74-26 *nrdJm* (TVNrdJm) was constructed with SwissModel (*Waterhouse et al., 2018*) using the *Lactobacillus leichmannii* monomeric NrdJ as template (PDB: 1L1L) (*Sintchak et al., 2002*) and default parameters.

## Recombinant gene expression and purification

Recombinant expression of the TV P74-26 *nrdJm* (UniprotKB: A7XXH5) was performed with a synthetic gene in a pET28b(+) expression vector and *E. coli* BL21(DE3) expression strain. 800 mL of LB-medium, containing 30 μg mL$^{-1}$ kanamycin, were inoculated from a preculture to a cell density of $OD_{600} = 0.1$ and incubated at 37 °C and 130 RPM. At an optical density of 1.0 expression was induced by addition of IPTG to a final concentration of 0.1 mmol L$^{-1}$. Expression was performed at 37 °C and 130 RPM for 4 h. After cell lysis with lysozyme and sonication, the enzyme was purified by Ni-NTA affinity chromatography followed by size exclusion chromatography.

*Thermus thermophilus* thioredoxin gene *trx1* (UniprotKB: Q72HU9) and the thioredoxin reductase *tr* gene (UniprotKB: Q72HD8) were cloned from genomic DNA in a pET52b(+) expression vector with C-terminal His-Tag. Expression was performed in a *E. coli* BL21(DE3) expression strain. For all genes, 800 mL of LB-medium, containing 100 μg mL$^{-1}$ ampicillin, were inoculated from a preculture to a cell density of $OD_{600} = 0.1$ and incubated at 37 °C and 130 RPM. At an optical density of 1.0 expression was induced by addition of IPTG to a final concentration of 0.1 μmol L$^{-1}$. Expression was performed at 130 RPM for 16 h at 37 °C for Trx1 and at 20 °C for TR and. After cell lysis with lysozyme and sonication, the enzyme was purified by Ni-NTA affinity chromatography followed by

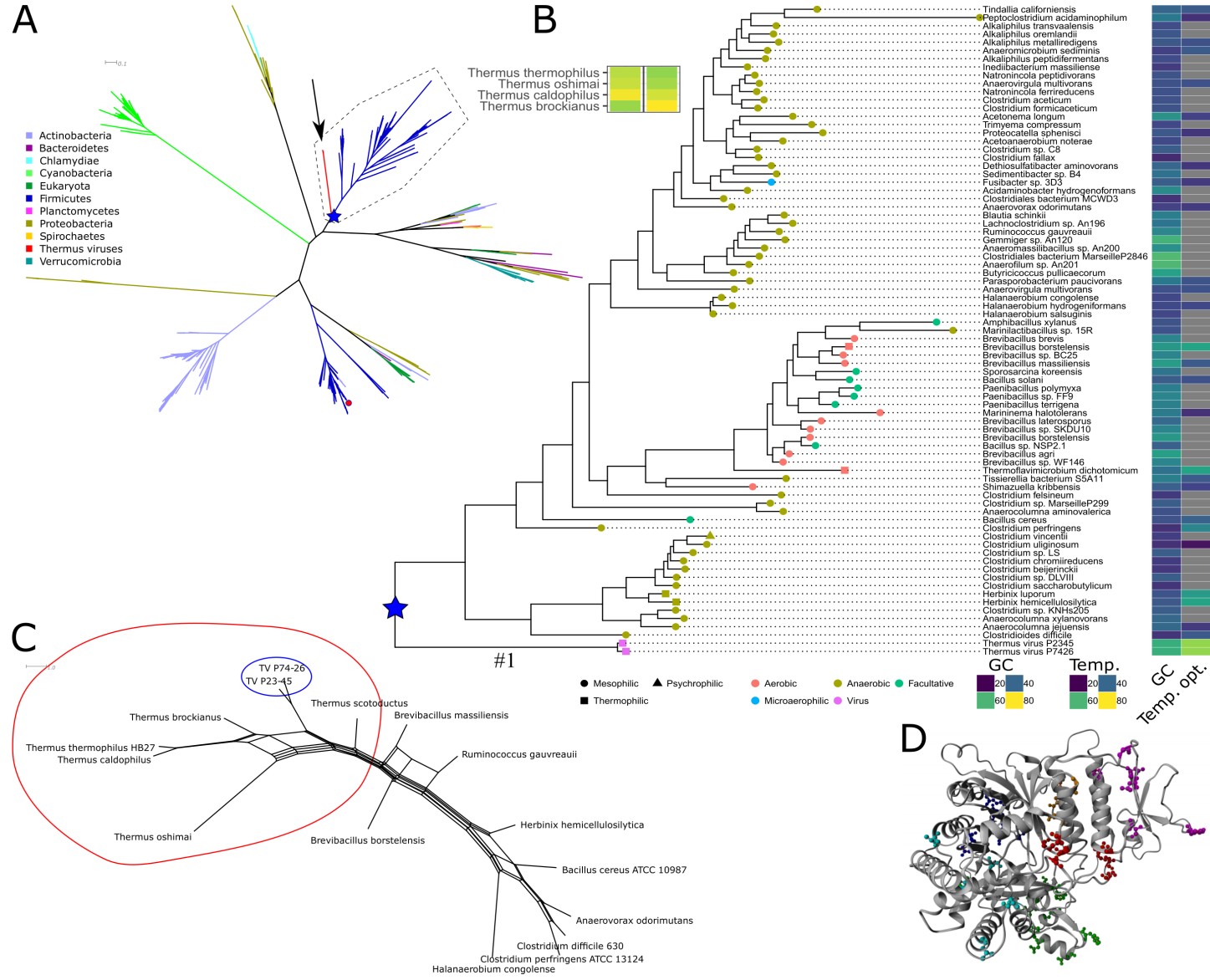

**Figure 1** **Phylogeny, sequence characteristics and growth optima.** (A) Unrooted maximum likelihood phylogeny of representative NrdJm amino acid sequences. The two virus sequences are indicated with an arrow. Phyla, with viruses assumed from their names to infect organisms from the same phylum, are indicated with coloured branches. The only solved structure from Lactobacillus leichmannii (PDB: 1L1L) is indicated with a small red circle. The root of the TV and Firmicutes clan is marked with a blue star. Full tree file available at Figshare (https://doi.org/10.17045/sthlmuni. 7117430.v1). (B) Detailed view of the clan containing the Thermus virus sequences, surrounded by a dashed line in A, with GC contents, optimum growth temperatures (when available; viruses set to typical host 70 °C), indications of temperature ranges (leaf node shape) and oxygen require- ments (node colour). (C) NeighborNet depicting patterns of codon usage between *Thermus thermophilus*, Thermus viruses and selected Firmicutes measured in euclidean distances. A split containing *Thermus* spp. and TVs is encircled in red; the TVs are encircled in blue. (D) Positions of muta- tions with a posterior probability ≥ 95% to be positively selected, mapped on a homology model of TVNrdJm: red: specificity site, magenta: dimer mimicking domain, orange: proximity of the specificity site, green: B12 binding site. Blue: surface patch, cyan: others.

desalting. Details of the purification procedure are given in the supporting information. An SDS-PAGE showing the purified NrdJm, Trx1 and TR is attached in the supporting information (Fig. S1).

Analytical size exclusion chromatography was performed on a Superdex$^{TM}$ 200 10/300 GL column. 500 μg of TV P74-26 NrdJm were loaded on the column and eluted with SEC buffer (50 mmol L$^{-1}$ tris–HCl, 300 mmol L$^{-1}$ NaCl, 1 mmol L$^{-1}$DTT, pH = 8.0) with a flow rate of 0.2 ml min$^{-1}$.

## Ribonucleotide reductase activity assays

If not stated differently, RNR activity was measured for the conversion of the substrate GTP to the corresponding product dGTP in the presence of a dTTP effector. The standard setup of the assay contains 50 mmol L$^{-1}$ Tris (pH = 8.0), 50 mmol L$^{-1}$ MgCl$_2$, 50 mmol L$^{-1}$ Dithiothreitol (DTT), 1 mmol L$^{-1}$ dTTP, 1 mmol L$^{-1}$ GTP, 10 μmol L$^{-1}$ AdoCbl and 0.1–0.4 μmol L$^{-1}$ NrdJm. In assays containing redoxin DTT was omitted and 2.5 μmol L$^{-1}$ Trx1, 1 μmol L$^{-1}$ TR and 2 mmol L$^{-1}$ NADPH were added. The assays were performed in 50 μL scale in three independent experiments for each condition. The assay was started by addition of the RNR and stopped after 5 to 15 min by addition of 50 μL Methanol and incubation at 50 °C for 5 min. After centrifugation and addition of 200 μL dH$_2$O, the samples were analyzed via UHPLC.

UHPLC analysis was performed on a Knauer Platinblue® UHPLC. 5 μL of sample were injected on a Luna Omega 1.6 um Polar C18® column from Phenomenex. GTP/dGTP and UTP/dUTP were eluted with a linear MeOH gradient from 26–30% over 15 min in a 50 mmol L$^{-1}$ KP$_i$-buffer (pH = 7.0) containing 0.25% (v/v) tetrabutylammonium hydroxide. CTP/dCTP were eluted with a linear MeOH gradient over 15 min from 27–30% the same buffer. ATP/dATP were eluted with a linear MeOH gradient over 15 min from 10–30% in the buffer as described before. A complete table of retention times is given in the supporting information (Table S17).

# RESULTS

## Gene identification and phylogenetic analysis

Screening the RNRdb (http://rnrdb.pfitmap.org) for viral RNRs with no similar host RNRs, we found three NrdJm enzymes, i.e., monomeric class II RNR enzymes, from viruses infecting *Thermus* species. The viruses belong to the *Siphoviridae* family and were found in hot biotopes. Thermus virus P74-26 and P23-45 were isolated at Uzon caldera in Kamchatka (*Minakhin et al., 2008*; *Yu, Slater & Ackermann, 2006*). The similar Thermus virus g20c was isolated at Geyzer Valley, Kamchatka (*Xu et al., 2017*). The NrdJm protein from Thermus viruses P74-26 (TVNrdJm, txid466052) and P23-45 (txid466051) have 96% identical amino acid sequences. Since the sequence from TV g20c contains a likely intein, we excluded this sequence from further study. The phylogenetically closest well characterized homologue, from which a crystal structure is available, is the NrdJm from *Lactobacillus leichmannii* (*Booker & Stubbe, 1993*; *Sintchak et al., 2002*). This is currently the only well studied NrdJm enzyme and is only distantly related to TVNrdJm (Fig. 1A). Within the phylum Deinococcus-Thermus no NrdJm enzymes could be found in current databases.

The most closely related enzyme found in *Thermus* spp. are dimeric NrdJ enzymes with no significant sequence similarity to the TV enzymes.

We investigated the evolutionary descent of the three viral nrdJm genes by phylogenetic analysis of their respective amino acid sequences by estimating a maximum likelihood phylogenetic tree of the full diversity of NrdJm sequences from NCBI's RefSeq database (Fig. 1A). The two viral sequences grouped inside a well-supported (82% bootstrap support) clan consisting exclusively of Firmicutes sequences except for the viral sequences (Fig. 1B). Searching for metagenomic sequences similar to those in the TVNrdJm clan, only resulted in sequences placed in the Firmicute part of the clan (https://doi.org/10.17045/sthlmuni.7642343.v1). The identity between the viral sequences and the Firmicutes sequences was 35–42% at the amino acid level. At nucleotide sequence level the similarity was hardly detectable using the BLAST algorithm. The similarity to genes from Firmicutes was shared with five other genes from the virus, whereas eleven genes were most similar to genes from Proteobacteria and eight from Deinococcus-Thermus (Table S8).

Since the NrdJm protein sequences are most closely related to Firmicutes sequences, we wanted to see if the same held true for the genetic signature of sequences. Therefore, we compared codon usage and GC-content of the TV mrdJm genes to *Thermus* and Firmicutes species. In codon usage, measured in euclidean distance, TV *nrdJm* genes showed high similarities to the genomes of the *Thermus* species and lower similarities to Firmicutes (Fig. 1C). Since the similarity between Firmicutes and the *Thermus* species on average is even lower, the TV *nrdJm* codon usage is intermediary between *Thermus* spp. and Firmicutes. The GC content of the viral genomes as well as the *nrdJm* genes was 57–58% (Fig. 1B), relatively close to the GC-content of the *Thermus* hosts at 67–69%. For the Firmicutes harboring the most similar *nrdJms*, the GC-content was, with the exception of three taxa around 60%, between 27 and 54% (Fig. 1B). *Thermus* spp. are thermophilic organisms with optimal growth temperatures between 60 and 70 °C. Most of the relevant Firmicutes are mesophiles, but some are at least moderate thermophiles despite their low GC content. *Herbinix hemicellulosilytica*, for instance, has an optimal growth temperature of 55 °C with a GC content of only 36% (*Koeck et al., 2015*) (Fig. 1B).

To investigate if positive selection had played a role in the evolution of the Thermus viruses since their divergence from the common ancestor with the Firmicute sequences, we performed a test of the ratio of non-synonymous to silent nucleotide substitutions (dN/dS) on this branch (marked with "#1" in Fig. 1B) with codonml from PAML (*Yang, 1997*; *Yang, 2007*). We found a dN/dS ratio of 17.5, significantly greater than 1 (*p*-value: 0.036) and hence suggesting positive selection for amino acid substitutions. A Bayes Empirical Bayes analysis discovered 40 amino acid positions with a ≥95% posterior probability of being under positive selection (Table S10). To gain some insight into the possible role of the 40 amino acids, we modeled the TVNrdJm on the *Lactobacillus leichmannii* monomeric NrdJm structure (*Sintchak et al., 2002*) (Model in the SI) and mapped the positions of the significant non conservative mutations on to the model. In total, 18 of these positions are located around the specificity site or on the dimer mimicking domain. Ten positions are

located in the $B_{12}$ binding domain and another five cluster in a patch on the surface of the protein (Fig. 1D).

## Recombinant expression of nrdJm genes and oligomeric state

The gene coding for TVNrdJm was synthesized commercially with codon optimization for recombinant expression in *E. coli*. High levels of soluble expression were obtained after only 4 h of expression at 37 °C. After purification with IMAC and SEC we obtained 40 mg of pure enzyme per liter of culture. The enzyme eluted after 16.8 mL on the size exclusion column, corresponding to a molecular weight of 65 kDa. With a calculated molecular weight of 73 kDa, this corresponds to a monomeric state of the enzyme in solution (Fig. S2).

## The viral *nrdJm* gene encodes a functional $B_{12}$-dependent ribonucleotide reductase with high temperature optimum

The purified TVNrdJm was able to reduce GTP to the corresponding deoxyribonucleotide but not GDP, defining it as a nucleoside triphosphate reductase. Equimolar concentrations of cofactor $B_{12}$ were sufficient to support the reaction, while higher concentrations reduced the activity to 40% at 0.1 mmol $L^{-1}$ (Fig. 2A). Without the presence of cofactor $B_{12}$, no reaction was observed.

As the enzyme was isolated from a virus with a thermophilic host, reaction temperatures were monitored over a temperature range from 25 °C to 90 °C. Enzyme activity peaks between 60 °C and 70 °C with complete inactivation at higher temperatures (Fig. 2B). The pH profile of the enzyme was investigated between pH values 5 and 9. The pH optimum was found to be between 7 and 8 with about 50% residual activity between 6.5 and 9 (Fig. 2C).

The allosteric regulation of TVNrdJm was investigated by a survey of the reduction of the four ribonucleotides ATP, CTP, GTP and UTP with the potential allosteric effectors dATP, dCTP, dGTP and dTTP. Without allosteric effector, no activity could be observed for any of the four substrates (Table 1). Significant activity was only present for the substrate/effector pairs ATP/dGTP, CTP/dATP, GTP/dTTP.

## Terminal reduction of NrdJm with artificial reducing agents and *Thermus* Thioredoxins

Activity of TVNrdJm was tested with the artificial reductants DTT and TCEP. While DTT is able to reduce the active site cysteine pair directly, TCEP can only work via the C-terminal cysteine pair (*Domkin & Chabes, 2014*; *Loderer et al., 2017*). The NrdJm exhibited 33% higher activity with TCEP compared to DTT at their respective maxima (Fig. 3A). For both reductants, the maximal activity was reached already at 2 mmol $L^{-1}$. Higher concentrations lead to a reduction of the activity to about 50% for both reductants at 100 mmol $L^{-1}$.

For the RNR catalyzed reaction *in vivo*, reducing equivalents are required in form of gluta- or thioredoxins. Since the TVs do not encode redoxins, we tested TVNrdJm with the thioredoxin/thioredoxin reductase system from the strain *Thermus thermophilus*. With the redoxin system, the viral NrdJm was active with a comparable reaction velocity to the artificial reductants (Fig. 3B). The control reaction without thioredoxin also displayed some activity. This may have been caused by residual DTT in the reaction mixture coming

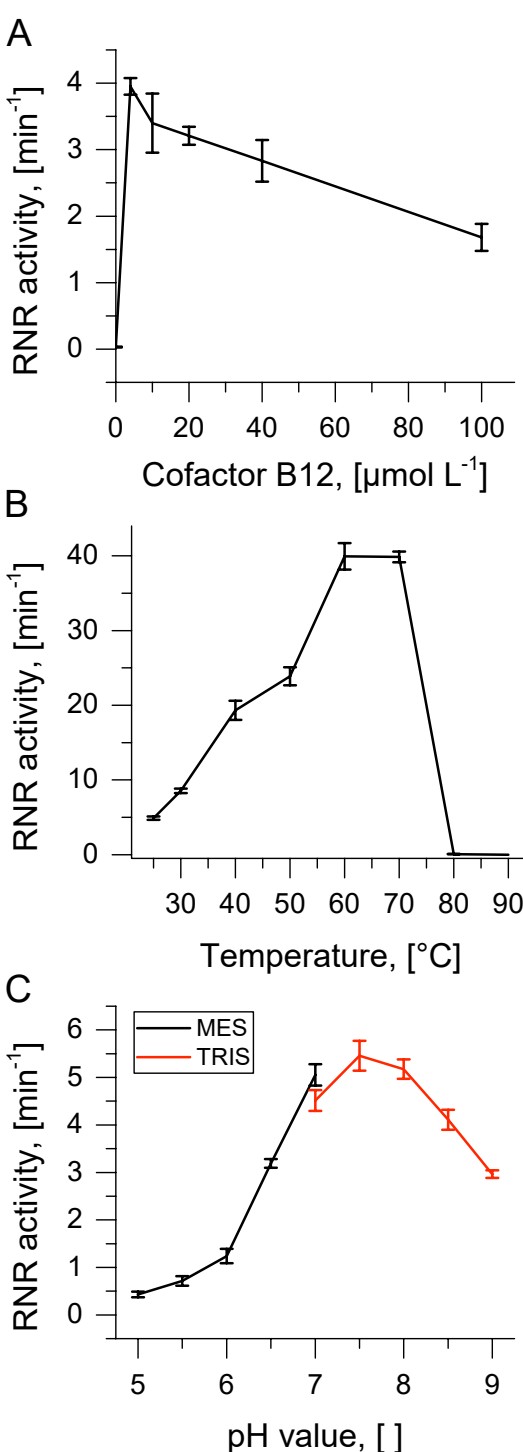

**Figure 2** **Biochemical characterization of TVNrdJm.** Effect of (A) B12-concentration (B) reaction temperature and (C) pH value on ribonucleotide reductase activity. Error bars represent the standard deviation of three independent experiments.

**Table 1** **Allosteric regulation of TVNrdJm.** RNR activity as kcat [min$^{-1}$] for the four potential native substrates with dNTP effectors. n.d.: not determined, n.s.: no separation of effector and reaction product via HPLC. Errors indicate the standard deviation of three independent experiments.

|  | dATP | dCTP | dGTP | dTTP |
|---|---|---|---|---|
| ATP | n. d. | 0.4 ± 0.1 | 2.2 ± 0.3 | 0.0 ± 0.0 |
| CTP | 0.8 ± 0.2 | n. d. | 0.0 ± 0.0 | 0.0 ± 0.0 |
| GTP | 0.1 ± 0.0 | 0.1 ± 0.1 | n.d. | 4.6 ± 0.2 |
| UTP | 0.2 ± 0.2 | n.s. | n.s. | 0.0 ± 0.0 |

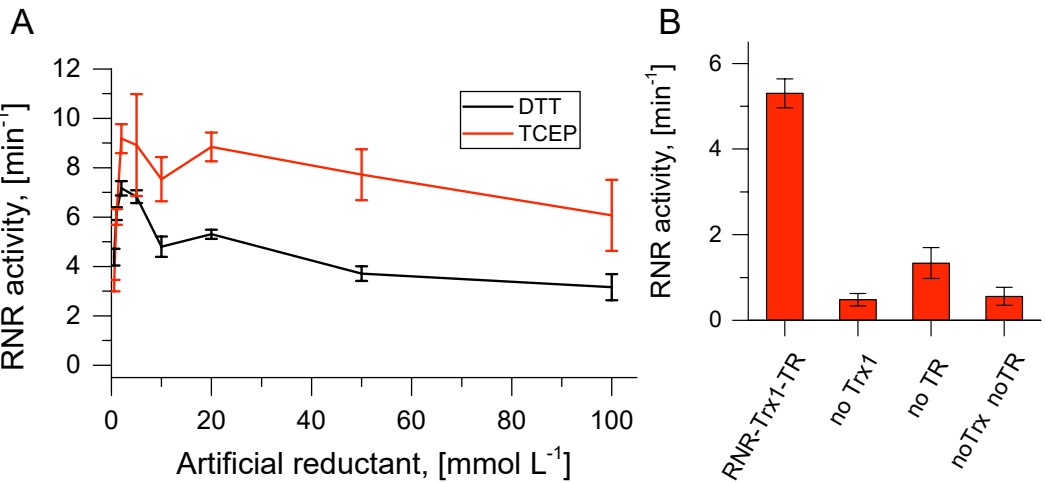

**Figure 3** **Artificial and potentially native reduction systems for NrdJm.** (A) Effect of the concentration of the artificial reductants DTT and TCEP on NrdJm activity. (B) NrdJm activity with thioredoxin Trx1 and thioredoxin reductase TR. Different control reactions were performed without thioredoxin and thioredoxin reductase. Error bars represent the standard deviation of three independent experiments.

from the enzyme stocks. The experiment also showed that the complete redox reaction chain from NADPH over thioredoxin reductase, thioredoxin and RNR to the final electron recipient GTP is functional in vitro.

## DISCUSSION

We discovered three viruses isolated from *Thermus* spp. that contain monomeric NrdJm RNRs, not found in their hosts or in related organisms. In the *Deinococcus-Thermus* phylum, no NrdJm enzymes could be found, opening the question of the evolutionary origin of the viral enzymes and their role in a potentially ongoing horizontal transfer of NrdJm RNRs by transduction. Phylogenetic analysis of amino acid sequences of NrdJm RNRs showed that the closest related sequences are found in Firmicutes, most of which are mesophilic although we also discovered a small number of moderate thermophiles. Further, five additional viral genes, dispersed throughout the Thermus virus P74-26 genome (Table S8), potentially share the same history as they were more similar to genes from Firmicutes than anything else in the RefSeq database. This relatively large number of genes with highest similarity to Firmicutes or prophages integrated in Firmicutes genomes, suggest a distant

evolutionary relation between the TV and Firmicutes. This could involve either a highly extended host range of the TV or one of its ancestors, or an evolutionary process where phages originally infecting Firmicutes (given the recruited NrdJm gene) have evolved a shift in host specificity. Given the limited host range seen for TV (Yu 2006) and that infectivity of viruses are typically limited even between strains of the same bacterial species and only a few phages have shown an ability to infect across genus (*Sullivan, Waterbury & Chisholm, 2003*) or class (*Jensen et al., 1998*) boundaries, it appears more likely that host shift played the most important role. Moreover, the large phylogenetic distance between the potential hosts, as well as a large number of TV genes most similar to genes in Proteobacteria and Actinobacteria (Table S8), suggests this was likely a lengthy process including intermediate hosts (*Hendrix, 2003*; *Hendrix et al., 1999*). It should be taken into consideration that TV could potentially have been able to retrieve the NrdJm gene from another intermediate host, but given that no sequences of higher similarity to the TV NrdJm has been detected, this cannot be verified.

The convergence of codon usage and GC content of the TVNrdJm gene and the *Thermus* genomes, show host adaptation of the gene at DNA sequence level. But adaptation also took place at amino acid sequence level, shown by the high dN/dS value between the branch leading to the Thermus virus compared with the rest of the TV/Firmicutes clan. The location of the 40 positions with significant positive selection hints at the traits, where selection took effect. The 18 amino acid substitutions with $\geq$95% posterior probability of being under positive selection, located in the proximity of the specificity site, are particularly suggestive as RNRs are responsible for the relative sizes of dNTP pools (*Hofer et al., 2012*; *Mathews, 2014*; *Nordlund & Reichard, 2006*). This could be a sign of adaptation to a genome of higher GC-content—close to 60% in TV compared with 25–52% in the Firmicutes' genomes—not only by its coding sequence, but also by how substrate specificity of the enzyme is regulated. Like all RNRs studied so far—except for RNRs in some members of the Herpesviridae family (*Averett et al., 1983*)—allosteric regulation of substrate specificity is intact in TVNrdJm and comparable to other described RNRs (Table 1) (*Eliasson et al., 1999*; *Larsson et al., 2004*; *Loderer et al., 2017*; *Rozman Grinberg et al., 2018*; *Sintchak et al., 2002*).

The TVNrdJm enzyme is a functional nucleoside triphosphate class II RNR, dependent on cofactor $B_{12}$. The monomeric state in solution as well as the sequence similarity to *L. leichmannii* NrdJm suggest that TVNrdJm contains a dimer mimicking domain, as described for *L. leichmannii* (*Sintchak et al., 2002*). Moreover, the enzyme shows an intact allosteric regulation comparable to other described RNRs (Table 1). To our knowledge, this is only the second NrdJm to be characterized thoroughly. Because of the degree of divergence between what appears to be monomeric NrdJms (Fig. 1A), our characterization of a second member of this subclass confirms what was earlier only assumed as shared features, such as the monomeric oligomeric state and the triphosphate specificity (*Lundin et al., 2015*; *Nordlund & Reichard, 2006*).

The enzyme has a temperature optimum at 60–70 °C, which is in the range of the optimal growth temperatures of most *Thermus* species (*Oshima & Imahori, 1974*; *Williams et al., 1996*). Although our in vitro assay conditions do not fully reflect the *in vivo* conditions, we

can still conclude that the required thermal stability is present. As potential native reducing equivalent, we found that Trx1 from *Thermus thermophilus* can provide electrons for the nucleotide reduction. This is different from *E. coli* T4 phage, where a viral Thioredoxin is mediating the electron transfer (*Berglund & Holmgren, 1975*). Hence, the viral NrdJm enzyme is capable of directly tapping the host's NADPH pool for deoxyribonucleotide biosynthesis. Since *Thermus* species also possess a biochemical pathway for vitamin $B_{12}$ biosynthesis, all biochemical requirements for the viral NrdJm to work in an infected *Thermus* cell are met (*Jiang et al., 2013*).

## CONCLUSIONS

Thermus virus P74-26 obtained a class II RNR from a phylogenetically distant source and adapted it for functional expression in the virus's host strains. Signs of the adaptation processes at DNA level were found in codon usage and GC content, while a high dN/dS value argues for positive selection at amino acid sequence level. Biochemical characterization revealed a functional, thermotolerant and monomeric RNR with the ability to retrieve electrons from its *Thermus* sp. host thioredoxin system. In contrast to the adaptation to the recent host, the details of its evolutionary descent remain enigmatic, limited by a lack of sequence information but also by our generally poor understanding of the evolution of viral host specificities.

Although not a completed transductive HGT yet, the mechanisms we describe in this article should be highly relevant for transduction events in general. The about 100-fold higher mutation rate of dsDNA viruses compared to bacteria could facilitate the adaptation process compared to that in the microorganism itself (*Lynch, 2010*; *Sanjuan et al., 2010*). Thus, the virus would not be a mere vector for DNA transfer but the genetic equivalent of a pressure chamber, where genes can become preadapted to a new environment. In our case, the described adaptation of an *nrdJm* gene most similar to Firmicutes, to work for the virus in the *Thermus* host, could act as preadaptation for a possible, albeit not yet observed, transfer to the host itself. This mechanism can be imagined for any gene that is able to deliver a direct selective advantage for the virus and would therefore not be restricted to RNRs.

## ACKNOWLEDGEMENTS

We kindly acknowledge Eugen Schell and Gerda Claus for their lab work contributions and Britt-Marie Sjöberg for discussions. We also thank two anonymous reviewers whose suggestions improved the manuscript.

### Funding

Christoph Loderer was supported by Fonds der Chemischen Industrie (FCI) (SK201/03) and funds of the Zukunftskonzept of TU Dresden (Federal and State Excellence Initiative). Daniel Lundin was supported by a grant (2016-01920) from the Swedish Science Research

Council to Britt-Marie Sjöberg. Karin Holmfeldt was supported by a grant (2013-4554) from the Swedish Research Council. The funders had no role in study design, data collection and analysis, decision to publish, or preparation of the manuscript.

### Grant Disclosures

The following grant information was disclosed by the authors:
Fonds der Chemischen Industrie (FCI): SK201/03.
Funds of the Zukunftskonzept of TU Dresden (Federal and State Excellence Initiative).
Swedish Science Research Council to Britt-Marie Sjöberg: 2016-01920.
Swedish Research Council: 2013-4554.

### Competing Interests

The authors declare there are no competing interests.

### Author Contributions

- Christoph Loderer, Karin Holmfeldt and Daniel Lundin conceived and designed the experiments, performed the experiments, analyzed the data, contributed reagents/materials/analysis tools, prepared figures and/or tables, authored or reviewed drafts of the paper, approved the final draft.

### Data Availability

Lundin, Daniel; Loderer, Christoph (2018): NrdJm phylogeny. figshare. Dataset. https://doi.org/10.17045/sthlmuni.7117430.v1.

Eren, A. Murat (2017): HBDs-AA-SEQUENCE-DATABASE. figshare. Fileset. https://doi.org/10.6084/m9.figshare.4902917.v1.

Lundin, Daniel; Loderer, Christoph; Holmfeldt, Karin (2019): Metagenomic NrdJm5 sequences placed in full phylogeny. figshare. Fileset. https://doi.org/10.17045/sthlmuni.7642343.v1.

Lundin, Daniel; Loderer, Christoph; Holmfeldt, Karin (2019): dN/dS calculation of Thermus virus NrdJm evolution. figshare. Fileset. https://doi.org/10.17045/sthlmuni.7642463.v1.

### Supplemental Information

Supplemental information for this article can be found online at http://dx.doi.org/10.7717/peerj.6700#supplemental-information.

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
