# Peer review of "Non-host class II ribonucleotide reductase in Thermus viruses: sequence adaptation and host interaction"

_PeerJ, doi:10.7717/peerj.6700_

## Round 0.1 · original submission · Major Revisions

Dear Dr. Loderer and colleagues:

Thanks for submitting your manuscript to PeerJ. I have now received three independent reviews of your work, and as you will see, two reviewers raised some concerns about the research. Nonetheless, there is plenty of all-around optimism for me to encourage you to revise your work and resubmit. I am sure that addressing these concerns will greatly improve your manuscript and bring it close to publication. Thus, I strongly encourage you to take into account all of the criticisms raised by the three reviewers.

In your revision, please consider including meta genomic sequences in your phylogeny estimations, or provide a reason why this is not necessary. Please also make sure the figures and tables are adequately described, and that they are stand-alone, but also that items mentioned in the text are easily garnered from the figures/tables. Please also make sure the all supplementary material is stand-alone. Be as descriptive as necessary. Please provide a rationale for tree-rooting. Please ensure that all models are adequately described, and also why certain criteria/parameters were selected. Finally, please be sure that all hypotheses for lateral gene transfer are strongly supported by the data (and discussed thoroughly).

Accordingly, I am recommending that you revise your manuscript, taking into account all of the issues raised by the reviewers. I look forward to seeing your revision, and thanks again for submitting your work to PeerJ.

Good luck with your revision,

-joe

Reviewer 1 ·

Basic reporting

Article is well written and constructed.

Experimental design

Experimental design is adequate. I would like to see inclusion of meta genomic sequences in phylogenetic analysis.

Validity of the findings

Overall I agree with the findings of this article and feel that the conclusions represent a potential interpretation of these data. I do raise a few questions in General comments that I would like to see addressed.

Additional comments

This manuscript describes the relationship of two Thermus Virus derived monomeric class II ribonucleotide reductases (NrdJm). In that NrdJm enzymes are perhaps the most undercharacterized type of Nrd’s, this manuscript’s phylogenetic and biochemical characterization of these enzymes is highly significant and a worthy contribution. As these viruses infect Thermus sp. which have never been shown to carry NrdJm, the authors are also able to speculate about the origin of the gene in these viruses and draw conclusions regarding the process of viral mediated HGT between taxa and the subsequent process of adaptation.

Overall I like this manuscript. It is well written. The biochemical characterizations are strong. The phylogenetic analysis is soundly performed; and a range of sequence and strain characteristics are presented to support the authors’ arguments. I see the authors’ supposition that the Thermus Virus gene was donated by a Firmicute to be largely a plausible explanation of the data. However, I do raise several questions that I would like to see the author’s address/consider before this manuscript is accepted. Importantly the exclusion of any metagenomic or other environmental sequences in the phylogenetic analysis (currently based upon RefSeq genomes only) may lead to false conclusions regarding the phylogenetic closest neighbors. Inclusion of any environmental uncultured microbial and viral sequences on this tree might substantially alter the topology — while not necessarily negating the common orgin of Firmicute and TV NrdJm, it may show that an intermediate group of bacterial or viral sequence space may exist that would alter some of the conclusions.

Specific comments:

Ln 219. Did the analysis allow the possibility that some of the genes in the genome might be more similar to other viruses than to bacterial taxa? How about similarity to metagenomic sequences (viral or microbial) or SAG's? Such sequences if they exist might substantially alter the phylogeny. Quite often viral genes are divergent enough that simply aligning sequences to known genomes can give quite a different picture than aligning to environmental sequences (i.e. they look similar to some group — firmicutes — only because they haven’t yet been put in proper context).

Figure 1a. While I understand the authors' intent of showing the larger world of NrdJm in this figure, however the fact that most of the leaves in this tree are outside of the Thermus viruses and Firmicutes phyla means that the phylogenetic relationships within these more relevant taxa are not as accessible for the reader. A more focused tree (perhaps an inset or supplement) might even give the possibility of adding a ring indicating the GC content or other metadata — I would really like to be able to see the phylogeny of the Firmicutes in relationship to their GC content. How well do these NrdJm phylogenies follow 16S phylogeny? This tree is showing a clear divergence of the viral sequences from all Firmicutes, making it seem that this is a fairly ancient split and that all these firmicute taxa diverged after the split. If this is the case, does the fact that TVNrdJm is still middle ground in its GC content relative to the Thermus host make sense? I can see a couple of other explanations: 1) this virus may infect other hosts; Thermus may not even be the optimal host. 2) The gene form as it exists in the TV’s has more recently transferred to the virus from another host that is not represented on this tree . . . perhaps completely unknown. These possibilities are partially addressed in the discussion 314-319, but I feel it could be further explored.

There is mention of five other genes most similar to Firmicutes (Ln 219). Are any of these adjacent to NrdJm? Might they have been introduced simultaneously? Do any of them show similar evidence of adaptation, codon usage shifts, etc?

Ln 207. How many genomes are sequenced in Deinococcus-Thermus? How many from similar environments to these viruses? HOw do these numbers compare to number of Firmicutes in the database? Extremophile Firmicutes?

Ln 116 - Are the identities/accession number of these 363 sequences listed somewhere in the article?

Ln 116 - I assume these are protein sequences (amino acid) . . . worth stating this explicitly.

Ln 119 and 121 - 363 sequences with 363 residues length seems like an unlikely convergence. Just making sure one of these numbers is not an error.

Ln 140-143 - What sequences were dN/dS applied to? Only the ones reported in Figure 1b?

Ln 312 - Does this indicate that there is a relationship between TV and Firmicutes phage . . . or might it indicate that TV can be Firmicutes phage?

Ln 320 - While I agree the proximity of these viruses makes encounters between viruses of tehse taxa possible, I don’t see how it supports the common ancestry?

Ln 223 & Figure 1c. Why were these 6 chosen for this analysis (or are those just the ones you decided to display)? There seem to be two closely related leaves to C. diff on Fig 1a tree, are these among those displayed?

Figure 1. There is a lot going on for 1 figure. Seemingly loose connections between panels.

Figure 1a. Labels for TV’s might be larger font, similar to the L. leichmannii. The notation “(see text)” in the legend doesn’t make sense to me, what am I supposed to be looking at?

Figure 1c. Accession numbers of the sequences should be noted somewhere in the manuscript.

Figure 1d. A scale bar would be a good addition to the Neighbor-Net. Purpose of red and blue circles should be in figure legend.

Figure 1e. Why were these individuals chosen for this analysis, do they represent the extremes of the Firmicutes and Thermus Phyla? Are the strain identities recorded somewhere in the MS? It might be good to add secondary x-axis categories indicating the phylum to clarify Firmicutes vs Thermus division

Figure 1f. Very hard to see the non-conserved residues. Is the figure meant to convey that these aa substitutions are having a structural effect? If so, this really isn’t illustrated without seeing a reference structure (reconstructed ancestor or other).

Table 1. Some indication of how conservative/non-conservative the change is might be helpful. For instance I see quite a few non-polar to non-polar changes here, granted some have significantly different side chains. In any case this table would seemingly be more useful to the reader if it was more informative than just listing the substitutions.

Ln 135. TVNrdJm not defined until ln 147.

·

Basic reporting

no comment

Experimental design

no comment

Validity of the findings

no comment

Additional comments

This is a very good article which is written in a professional manner.
Experimental design is proper and all conclusions drawn by the authors are supported by the data presented.

Please check that all references for Figures and Tables made in the text are correct.
For example, in line 248 the authors mention Figure 1E but I believe they mean Figure 1F

Reviewer 3 ·

Basic reporting

This manuscript describes an interesting case of a viral gene that has been transferred from a different phylum.

The manuscript is generally well written but some sentences especially in the Discussion can be written more clear (e.g. sentences starting in lines 309, 311, 370).

Some more information can be given in the supplementary material, e.g., legends of Tables S3 and S4 are not self-explanatory. In SI_Figure_1B more information would be helpful, like the best e-value, the coverage of the hit and the sequence ID of the hit.

Table 1 is presented in a confusing way. It seems to me that column 1 and 2 are not related with column 3 and 4. It would be more clear to present a table with 2 columns and 13 lines.

Experimental design

The authors find “three NRdJm enzymes” (line 198), however later only two of them are described in detail (e.g., Fig. 1A). Please describe the rationale of focussing on these sequences.

A rooted cladogram is presented in Fig. 1A. Information on how the root was determined must be given. In addition, a cladogram might not be the appropriate representation here. I looked at the data using the “radial phylogram” option in Dendroscope which is more informative, in particular, the branch lengths are in substitutions per site as estimated with RAxML and are informative about evolutionary distances.

There is no sufficient information given on the dN/dS calculation (line 142). How was the Yang and Nielsen model applied, have the parameters been estimated with ML, has a program been used? In contrast to pairwise comparisons, PAML (doi:10.1093/molbev/msm088) also implements models along the phylogeny where the difference of dN/dS on specific branches can be tested. This would be more informative as the dN/dS of the ancestral branch and of the branches leading to TV could directly be compared.

Validity of the findings

The authors state in the introduction “HGT by transduction is a stepwise process in which the transferred genes need to be functional and provide a selective advantage first in the virus, then in the receiving cellular organism.” (line 66). I do not agree with this statement as, for example, also genes that are neutral for the virus could be transferred. The gene should certainly not be too harmful as explained in Soucy et al. (2015) but selective advantage is not a strict requirement.

The finding that the genes were transferred from a member of the Firmicutes to TV (as stated in line 307) is not supported as the ancestor of the TV sequences actually branches before the ancestor of Firmicutes. This is correctly stated in other parts of the paper and thus should be also mentioned correctly in the abstract and in the discussion.

The presentation of the non-conservative substitutions is confusing, in line 247, 76 are mentioned but later there are only 13 (line 332). It seems like these are only the one in the substrate specificity site and its vicinity but then it should be clearly stated (e.g., line 332).

---

## Round 0.2 · accepted · Accept

Dear Dr. Loderer and colleagues:

Thanks for revising your manuscript based on the concerns raised by the reviewers. I now believe that your manuscript is suitable for publication. Congratulations! There are few minor items to address, per reviewers 3. Please handle these before sending your manuscript to production. I look forward to seeing this work in print, and I anticipate it being an important resource for the communities studying non-host class II ribonucleotide reductase in viruses as well as other organisms. Thanks again for choosing PeerJ to publish such important work.

Best,

-joe

# Reviewer 1 ·

Basic reporting

The MS is well written.

Experimental design

Experiments are well designed.

Validity of the findings

The findings reported are well-supported and conclusions well-founded.

Additional comments

This is a resubmission of the version that I have previously reviewed. I have reviewed the response to review and revisions. I believe the authors have done a thorough job of addressing the previous comments and find the newly revised manuscript to be improved. I have no further recommendations.

·

Basic reporting

no comment

Experimental design

no comment

Validity of the findings

no comment

Additional comments

In my opinion, the authors have addressed all comments made by the reviewers

Reviewer 3 ·

Basic reporting

I am pleased to this this substantially improved manuscript. I have only few minor comments.

line 152: Why are some gene sequences missing? Every protein should have at least one gene sequence.

line 238: I suggest to change half to part.

line 265: How is the p-value calculated? Maybe add this information the the methods.

line 351: gen -> gene

line 352: genomes, show -> genomes shows

Experimental design

no comment

Validity of the findings

no comment